# The Lipid Status in Patients with Ulcerative Colitis: Sphingolipids are Disease-Dependent Regulated

**DOI:** 10.3390/jcm8070971

**Published:** 2019-07-04

**Authors:** Sarah Bazarganipour, Johannes Hausmann, Stephanie Oertel, Khadija El-Hindi, Sebastian Brachtendorf, Irina Blumenstein, Alica Kubesch, Kathrin Sprinzl, Kerstin Birod, Lisa Hahnefeld, Sandra Trautmann, Dominique Thomas, Eva Herrmann, Gerd Geisslinger, Susanne Schiffmann, Sabine Grösch

**Affiliations:** 1Institute of Clinical Pharmacology, Goethe University Hospital, Theodor-Stern-Kai 7, 60590 Frankfurt/Main, Germany; 2First Department of Internal Medicine, Division of Gastroenterology, Goethe University Hospital, Theodor-Stern-Kai 7, 60590 Frankfurt/Main, Germany; 3Institute for Biostatistics and Mathematical Modelling, Goethe University, Theodor-Stern-Kai 7, 60590 Frankfurt/Main, Germany; 4Fraunhofer Institute for Molecular Biology and Applied Ecology IME, Branch for Translational Medicine and Pharmacology TMP, Theodor-Stern-Kai 7, 60596 Frankfurt/Main, Germany

**Keywords:** ceramide, sphingolipid, ulcerative colitis, LC–MS/MS, S1P, EPA, DHA, patient

## Abstract

The factors that contribute to the development of ulcerative colitis (UC), are still not fully identified. Disruption of the colon barrier is one of the first events leading to invasion of bacteria and activation of the immune system. The colon barrier is strongly influenced by sphingolipids. Sphingolipids impact cell–cell contacts and function as second messengers. We collected blood and colon tissue samples from UC patients and healthy controls and investigated the sphingolipids and other lipids by LC-MS/MS or LC-QTOFMS. The expression of enzymes of the sphingolipid pathway were determined by RT-PCR and immunohistochemistry. In inflamed colon tissue, the de novo-synthesis of sphingolipids is reduced, whereas lactosylceramides are increased. Reduction of dihydroceramides was due to posttranslational inhibition rather than altered serine palmitoyl transferase or ceramide synthase expression in inflamed colon tissue. Furthermore, in human plasma from UC-patients, several sphinglipids change significantly in comparison to healthy controls. Beside sphingolipids free fatty acids, lysophosphatidylcholines and triglycerides changed significantly in the blood of colitis patients dependent on the disease severity. Our data indicate that detraction of the sphingolipid de novo synthesis in colon tissue might be an important trigger for UC. Several lipids changed significantly in the blood, which might be used as biomarkers for disease control; however, diet-related variabilities need to be considered.

## 1. Introduction

Ulcerative colitis (UC) is a chronic disease characterized by diffuse inflammation of the rectal and colonic mucosa. The occurrence of UC has increased worldwide over the past few years [1,2,3]. The highest incidence of ulcerative colitis has been reported in Northern Europe (24.3 per 100,000) [3]. Patients most commonly complain about increased frequency and often bloody stools, abdominal pain, weight loss and general reduced quality of life. The clinical course is characterized by periods of remission and exacerbation, which may occur either spontaneously or in response to medical treatment. The primary aim of medical management is to induce and maintain remission with the long-term goals of preventing disability, colectomy, and colorectal cancer. Targets for remission include resolution of clinical symptoms, defined as cessation of rectal bleeding and improvement in bowel habits, and endoscopic healing, which is frequently defined as an endoscopic Mayo score of zero or one [4,5]. The selection of medications is guided by disease severity and extent. Mesalamine is a trusted standby for patients with mild-to-moderate UC. In cases of more severe inflammation, the use of systemic steroids, immunosuppressants (azathioprine and 6-mercaptopurine) and biologicals (antibodies against tumor necrosis factor alpha (TNFα) or integrin α_4_β_7_) is often warranted for disease control and to maintain remission. Several guidelines [6,7] are available to assist the physician in making his decision. However, use of these drugs is not without immediate and delayed risks such as infections, allergic reactions, development of psoriasis and an increased risk of malignancies. Thus, research has been focused on developing targeted immunomodulatory drugs aiming to potentially address colonic inflammation whilst avoiding a general alteration of immune response. One promising approach is the inhibition of the sphingosine-1-phosphate (S1P) signaling pathway by interfering with S1P receptors (S1PR). S1P signaling regulates inflammation via its impact on the trafficking, differentiation, and effector functions of bone-marrow-derived immune cells [8]. In the treatment of multiple sclerosis, S1PR agonists (like fingolimod) have already been applied and show enticing results [9]. Fingolimod operates through internalizing and consecutively degrading the S1PR1 in lymphocytes, thus enabling immune cells to leave secondary lymph nodes in the blood stream [10,11,12]. Phase III clinical trials using a S1PR1/5 agonist (ozanimod) for UC are currently ongoing and phase I and II results appeared promising [13]. S1P belongs to the group of sphingolipids which play a distinct role as a second messenger in signaling pathways and are essential for physiological properties of cell membranes [14,15]. Beside S1P, several other sphingolipids have been shown to be deregulated in various diseases, especially under inflammatory conditions [12,16,17,18,19]. Ceramides and their complex derivatives are main components of membranes and their deregulation impacts cell signaling and the fate of the cell [20]. Sphingolipids are synthesized either by the de novo synthesis or the salvage pathway. The serine palmitoyl transferase (SPT) is the first rate-limiting enzyme in the de novo synthesis, catalyzing the condensation of serine and palmitoyl-CoA to sphinganine. Six homologous ceramide synthases (CerS) attach various acyl-CoA side chains to sphinganine or sphingosine, generating either dihydroceramide (dhCer) or ceramide (Cer). They have chain length preferences using C14- to C26-acylCoAs. CerS1 mainly synthesizes C18-Cer, CerS4 synthesizes C18-/C20-Cer, CerS5 and CerS6 mainly synthesize C14- and C16-Cer, CerS2 synthesizes the long-chain C22/C24-Cer, and CerS3 synthesizes long ceramides with a chain length of up to C34 [21,22]. CerS are expressed tissue-specific and are regulated at transcriptional as well as posttranscriptional levels [23]. Ceramides are converted in the Golgi apparatus either to hexyl-ceramides or sphingomyelins, which are main components of cellular membranes. In the salvage pathway, complex sphingolipids are degraded back to ceramides and sphingosine, which could be phosphorylated to S1P and ceramide-1-phosphate (C1P). Recently published studies from human patients suffering from inflammatory bowel disease (UC or Crohn’s disease (CD)) give the first hint that the sphingolipid rheostat is deregulated in UC, however, the data are either generated from stool probes [24] or in untreated patients [25,26]. Furthermore, findings in various mice deficient of an enzyme in the sphingolipid pathway, indicate that sphingolipids play an important role in the development of ulcerative colitis [27,28,29,30,31,32,33,34,35]. These data indicate that the sphingolipid pathway, and especially the balance between different sphingolipids, may impact the development or progression of colitis. Against the background of this knowledge, we investigated the sphingolipid status and other lipids in human plasma, white blood cells and colon biopsies from UC patients. We analyzed whether differences in the sphingolipid status could be observed depending on disease severity and medical treatment in UC patients and whether lipids in the blood might be determined as biomarkers for disease control.

## 2. Experimental Section

### 2.1. Cohort

We included 98 patients who applied to the IBD outpatient clinic of the University Hospital Frankfurt with diagnosed ulcerative colitis between 2014 and 2018 in this prospective study. Inclusion criteria were age above 18 years, diagnosed ulcerative colitis, as well as written informed consent to participate in the study. Exclusion criteria were an age below 18 years, no definitive diagnosis of UC and underlying liver diseases. The patients were followed up at the treating physician’s discretion, usually every 6–12 weeks. In the course of the observation time, repeated blood samples were obtained from most patients. As some patients changed their disease stage during this time, we could compare intra- and inter-individual changes in our dataset (see also Appendix A). Blood samples were obtained during clinically indicated routine laboratory diagnostics and kept on ice for 1–4 hours prior to further analysis. Clinical parameters such as partial Mayo score (Appendix A), duration and localization of disease, and UC medication were assessed during patient visits (see Table 1 and Table 2). Furthermore, colon samples were collected from 59 European patients undergoing diagnostic colonoscopy at the department of Gastroenterology of the University Hospital Frankfurt. Biopsies were obtained from inflamed and non-inflamed sections of the colon. Non-inflamed control samples were taken from the same patients 2–5 cm above the location where samples of inflamed tissue were biopsied. All tissue samples were freshly frozen and stored at −80 °C immediately after resection. Furthermore, 148 control blood samples were taken from 25 healthy volunteers—repeated blood draw within one year, according to the patient samples—with a mean age of 40.9 years and kept on ice for 1–4 hours prior to further analysis. The study was conducted according to the declaration of Helsinki, and approval was obtained from the local ethics committee. Reference number: 296/14 and 77/15. A written informed consent was obtained from patients and volunteers prior to enrollment in the study.

### 2.2. Isolation of White Blood Cells (WBCs) from Blood Samples

About 9 mL whole-blood samples from UC patients or controls were collected in EDTA blood collection tubes and kept on ice for 1–4 hours until isolation of white blood cells (WBCs). For WBC-isolation, blood was centrifuged for 10 min at 2000× g at 4 °C, 1 mL plasma was taken and stored in aliquots at −80 °C for further analysis by LC–MS/MS. A quantity of 1 mL of PBS phosphate-buffered saline was added to the blood sample collection tube containing WBCs, erythrocytes, and plasma inverted several times and incubated with 45 mL erythrocyte-lysis buffer (135 mM NH_4_Cl; 10 mM NaHCO_3_; 0,1 mM Na-EDTA) for 10 min at RT and centrifuged at 600× g for 10 min. The WBC cell pellet was incubated again with 1 mL erythrocyte-lysis buffer for an additional 5 min, centrifuged at 300× g for 5 min and washed before being stored at −80 °C or directly used for RNA extraction. 

### 2.3. Real-Time qPCR

The mRNA from colon biopsies was extracted with the RNeasy Lipid Tissue Mini Kit (Quiagen, Hilden, Germany) according to the manufacturer’s instructions. About 2–5 mg of colon tissue was homogenized in 1 mL QIAzol lysis reagent, incubated for 5 min at RT before 200 µL chloroform was added. After centrifugation (15 min, 12,000 g, 4 °C), the upper phase (containing RNA) was applied to an RNeasy spin column, here the total RNA/DNA binds to the membrane. DNA was removed using DNase I and RNA was eluted in 30 µL of RNase-free water. RNA from white blood cells (WBCs) was isolated using the RNeasy Mini kit (Quiagen, Hilden, Germany) according to the manufacturer’s instructions. RNA content from tissue or WBC was determined using the NanoQuant Plate^TM^ and the Tecan plate reader (Tecan, Männedorf, Switzerland). The cDNA synthesis was performed using the Verso™ cDNA-Synthesis kit (Thermo Scientific, Schwerte, Germany), including an enhancer enzyme. The expression levels of CerS1, CerS2, CerS3, CerS4, CerS5, CerS6, Sptlc2, Sptlc3 and GAPDH were determined using the Syber Select Master Mix (Thermo Fisher Scientific, Darmstadt, Germany) with an ABI Prism 7500 Sequence Detection System (Applied Biosystems, Thermo Fisher, Damstadt, Germany). Relative mRNA expression was determined using the comparative ΔΔCT (cycle threshold) method, normalizing relative values to the expression level of human GAPDH. The sequences for the primer sets are shown in Table 1. Linearity of the assays was determined by serial dilutions of the templates for each primer set separately.

### 2.4. Liquid Chromatography Tandem Mass Spectrometry (LC-MS/MS) Analysis of Sphingolipids

Quantification of sphingolipids from tissue and plasma samples was performed by high-performance liquid chromatography tandem mass spectrometry. For quantification of sphingolipids, the tissue samples were first mixed with water and homogenized to a suspension of 0.05 mg/µL tissue using a swing mill (Retsch, Haan, Germany) with four zirconium oxide grinding balls for each sample (25 Hz for 2.5 minutes).

A quantity of 40 µL of the tissue suspension (in total 2 mg tissue) was mixed with 160 µL water, or 10 µL plasma was mixed with 190 µL water, together with 200 µL extraction buffer (citric acid 30 mM, disodium hydrogen phosphate 40 mM) and 20 µL of the internal standard solution containing sphingosine-d7, sphinganine-d7, sphingosine-1-phosphate-d7, C17-Cer, C16-Cer-d7, C18-Cer-d3, C24-Cer-d4, C17-LacCer, C18-dhCer-d3, C16-LacCer-d3 and C18-GlcCer-d5. The mixture was extracted twice with 600 µL methanol/chloroform/hydrochloric acid (15:83:2, v/v/v). The collected lower organic phases were evaporated at 45 °C under a gentle stream of nitrogen and reconstituted in 100 µL of tetrahydrofuran/water (9:1, v/v) with 0.2% formic acid and 10 mM ammonium formate. Afterwards, amounts of sphingosine, sphingosine-1-phosphate, sphinganine, sphinganine-1-phosphate, C16:0-Cer, C18:0-Cer, C18:1-Cer, C20:0-Cer, C24:0-Cer, C24:1-Cer, C16:0-GlcCer, C16:0-dhCer, C18:0-dhCer, C24:1-dhCer were analyzed by LC–MS/MS. A Luna C8 column (150 mm × 2 mm ID, 3 μM particle size, 100 Å pore size; Phenomenex, Aschaffenburg, Germany) was used for chromatographic separation. The HPLC mobile phases consisted of water with 0.2% formic acid and 2 mM ammonium formate (A) and acetonitrile/isopropanol/acetone (50:30:20, v/v/v) with 0.2% formic acid (B). For separation, a gradient program was used at a flow rate of 0.3 mL/min. The initial buffer composition 55% (A)/45% (B) was held for 0.7 min and then, within 4.0 min, linearly changed to 0% (A)/100% (B) and held for 13.3 min. Subsequently, the composition was linearly changed within 1.0 min to 75% (A)/25% (B) and then held for another 2.0 min. The running time was 21 min and the injection volume was 15 μL. After every sample, the sample solvent was injected for washing the column with a 12 min run. The MS/MS analyses were performed using a triple quadrupole mass spectrometer API4000 (Sciex, Darmstadt, Germany) equipped with an Electrospray Ionization (ESI) ion source. The analysis was performed in Multiple Reaction Monitoring (MRM) mode with a dwell time of 25 ms.

Data acquisition was carried out using Analyst Software V 1.6 and quantification was performed with MultiQuant Software V 3.0 (both Sciex, Darmstadt, Germany), employing the internal standard method (isotope dilution mass spectrometry). Variations in the accuracy of the calibration standards were less than 15% over the whole range of calibration, except for the lower limit of quantification, where a variation in accuracy of 20% was accepted.

### 2.5. Liquid Chromatography Quadrupole-time-of-flight Mass Spectrometry (LC-QTOFMS) Analysis of Lipids in Plasma

Plasma samples (20 µL) were extracted using methyl-tert-butyl-ether [1]. After re-extraction, the combined organic phases were separated into two aliquots. After drying under a nitrogen stream at 45 °C and stored at −40 °C pending analysis. The aliquots for negative ion mode were reconstituted in 60 µL methanol and those for positive ion mode in 60 µL of methanol/chloroform (2:1, v/v). LC–MS analysis was performed on a Nexera X2 system (Shimadzu Corporation, Kyoto, Japan) coupled with a TripleTOF 6600 with a DuoSpray ion source (Sciex, Darmstadt, Germany). The chromatographic separation was done on a Zorbax RRHD Eclipse Plus C8 1.8 µM 50 × 2.1 mm ID column (Agilent, Waldbronn, Germany) with a SecurityGuard Ultra C8 pre-column (Phenomenex, Aschaffenburg, Germany), using a binary gradient with 40 °C column temperature and a flow rate of 0.3 mL/min. The mobile phase A consisted of 10 mM ammonium formate and 0.1% formic acid in water and mobile phase B consisted of 0.1% formic acid in acetonitrile: isopropanol (2:3, v/v). The initial buffer composition with 20% B was held for 0.3 min, increased to 80% after 3.2 min, further increased to 100% after 7.0 min, held 100% B for 2.5 min and reverted to 20% B after 1.5 min, following a 2.5 min equilibration. For the negative measurement, 1 mM ammonium formate and 0.1% formic acid in water was used as mobile phase A. The run time was 17 minutes per polarity and the injection volumes were 2 µL for positive and 5 µL for negative ionization mode, respectively.

The MS analysis consisted of a TOF MS Scan from 100–1000 *m/z* with 250 ms of accumulation time and six data-dependent acquisitions per cycle with a mass range of 50–1000 *m/z* and 50 ms of accumulation time. Data acquisition was performed using Analyst TF 1.7 software and peak integration for semi-targeted comparison was done using MultiQuant V 3.02 (both Sciex, Darmstadt, Germany). The identification of the lipid species was based on the exact mass (+/- 5 ppm), the isotope ratio and the comparison of the MS/MS spectra with the reference spectra according to LIPID MAPS (http://www.lipidmaps.org) or METLIN (http://metlin.scripps.edu). Peak areas were normalized to the first quality control sample using median peak ratios by MarkerView 1.2 software (Sciex, Darmstadt, Germany) with a mass accuracy window of 10 ppm and 0.15 minutes retention time tolerance window for peak alignment. By normalizing with the median peak ratio, samples were corrected with a scale factor calculated from the median of the peak area ratio for each sample and a set reference sample for all peaks with a peak area greater than 1% of the largest peak of the reference sample. This method is more robust against asymmetrical intensity increase than the total intensity or total area sum normalization and hence can also be utilized for samples with changes in the content of more abundant lipids.

All samples were randomized prior to extraction. Pooled quality control samples were injected four times at the start and at the end of the run and after every ten samples to verify system stability.

### 2.6. Immunohistochemistry

CerS protein expression was detected in colon tissue biopsies.

For immunohistochemistry, 10 µM tissue sections were placed in 100% ice-cold methanol for 10 min and subsequently, in 100% acetone for one minute. Slices were washed and permeabilized in PBS containing 0.025% Triton X-100 for 2 × 5 min, then blocked in PBS containing 3% bovine serum albumin and 10% normal goat serum for 90 min at room temperature. The sections were incubated with anti-CerS3 (Sigma) primary antibody at 4 °C overnight, followed by Cy3 anti-rabbit (Sigma) secondary antibodies diluted 1:800 for 2 h in PBS containing 1% bovine serum albumin and 1% normal goat serum. The tissue was stained with DAPI (4′,6-Diamidine-2′-phenylindole dihydrochloride) (1:1000) (Applichem) for 10 min before being embedded into Aqua-Poly/Mount. Fluorescent measurements were done with the Axio Imager Z1 (Zeiss, Göttingen, Germany) with 20 and 63-fold magnification. 

### 2.7. Statistics

Sphingolipid levels and mRNA are presented as mean ± SEM (standard error of the mean). Statistical analyses were performed with GraphPad Prism7 software or R. Significant differences between groups were assessed using one-way ANOVA for three or more groups or two-tailed, two-sided Student’s t-tests for two groups. In case of significant ANOVAs, groups were mutually compared with t-tests by employing a correction of alpha according to Tukey.

## 3. Results

### 3.1. Patient Data

We collected 183 blood samples and 59 colon biopsies from 97 ulcerative colitis (UC) patients during a routine check-up at the department of gastroenterology at the Goethe University Frankfurt. Additionally, we collected 148 blood samples from 25 controls (healthy volunteers) within one year. According to the Mayo score, UC patients are divided into three groups (remission, mild, moderate/severe) (Table 2). Severity of disease and medication was taken from the electronic health report. All patients were aware of and consented to the study protocol. 

Disease duration and medication of ulcerative colitis (UC) patients is shown in Figure 1A,B. The majority of patients in this study suffered from colitis for more than five years. Most abundantly prescribed medication are aminosalicylates (like 5-ASA), anti-TNFα antibodies and glucocorticoids, either as single treatment but also often used in combinations.

### 3.2. Sphingolipids in Colon Tissue

We examined the sphingolipid status in colon tissue from UC patients using biopsies that were taken during a routine colonoscopy. From each patient, biopsies were taken from inflamed tissue and from tissue that showed no inflammatory symptoms (control). One of the most conspicuous results was that in inflamed tissue, the levels of sphinganine (dhSph) and most dihydroceramides (dhCer) decreased significantly (Figure 2A,B), indicating that in inflamed tissue, the sphingolipid de novo synthesis is reduced. No significant differences could be observed in the levels of sphingosine (Sph), ceramide (Cer) or glucosylceramide (GlcCer) between inflamed or control tissue (Figure 2A,C–E). Instead, the concentrations of C16:0- and C24:0-lactosyl-ceramide (LacCer) increased in inflamed tissue in comparison to control tissue (Figure 2F). Sphingosine-1-phosphate (S1P) and sphinganine-1-phosphate (SA1P) concentrations could not be quantified due to their low amount and the limited tissue mass.

Because most of the dihydroceramides decreased, we investigated the expression level of important enzymes of the sphingolipid de novo synthesis. The expression of serine palmitoyltransferase (SPT) and the six ceramide synthases (CerS) were investigated on mRNA levels in control and inflamed colon tissue. SPT is a heterodimer, which consists of two subunits Sptlc1 and Sptlc2 or Sptlc1 and Sptlc3. For mRNA expression, we determined the expression level of Sptlc2 and Sptlc3. Interestingly, there were no significant changes in the expression level of Sptlc2 or 3 nor of the six CerS but CerS1 and CerS3 show a tendency to increase in inflamed tissue compared to control tissue (Figure 3A). CerS3 showed a 12–20-fold increase in mRNA expression in inflamed tissue of several patients in comparison to matched control tissue. Therefore, we detected CerS3 protein by immunohistochemical (IHC) staining of colon tissue. IHC staining of CerS3 in control and inflamed colon tissue showed an enhanced expression of CerS3 in inflamed tissue, whereby most CerS3 staining could be observed in the lamina propria region (Figure 3B), which let us suppose that the increase in CerS3 expression is a result of invaded immune cells, but further studies have to proof this assumption.

### 3.3. Lipids in Blood

#### 3.3.1. Sphingolipids in Blood 

To investigate whether blood sphingolipid and other lipid levels in UC patients are disease –dependent regulated, we determined the sphingolipid concentrations in plasma from UC patients and controls (healthy volunteers) via LC–MS/MS and relative concentration of other lipids by LC-QTOFMS. The sphingolipid concentrations are shown to be dependent of the disease stage of the UC patients (Figure 4). The level of Sph, dhSph, Cer, GlcCer and most LacCer increased in blood of UC patients in comparison to control with a tendency to be most pronounced in patients who suffer from a moderate/severe disease stage. The concentration of C24:1-LacCer significantly decreased in the severe stage. Furthermore, the phosphorylated bases S1P and SA1P decreased in all stages of UC patients in comparison to control (Figure 4). The level of dhCer did not change between the different stages (data not shown). Disease stage-dependent changes in sphingolipid concentrations are also evident in single patients who underwent different stages over time (Appendix A). Pearson correlation analysis revealed that there are several highly significant correlations between the different sphingolipids in all patients which are shown in Appendix A. For example, there is a medium-to-strong linear positive interrelation between different long-chain ceramides, between distinct Cer and dhCer, Cer and GlcCer, different GlcCer, and C18:0-GlcCer and -LacCer (Appendix A). These data indicate that the different sphingolipids in blood are in a defined ratio to each other and that they significantly change in UC patients, which renders them interesting as possible biomarkers for disease control.

#### 3.3.2. Blood Sphingolipids in Correlation to Treatment with Aminosalicylate or Anti-TNFa

To investigate if the sphingolipid status is influenced by the drugs used for the treatment of patients in our cohort, we looked for the sphingolipid levels in patients treated either with monotherapy aminosalicylate or anti-TNF-α antibody in comparison to patients who received no medication. The disease scores of the patients treated either with aminosalicylate only or anti-TNF-α antibody are comparable (aminosalicylate: 64% remission, 26% mild, 10% moderate/severe; anti-TNF-α antibody: 65% remission, 20% mild, 15% moderate/severe; no treatment (none): 100% remission). Only C18:0-Cer significantly increased in patients treated with monotherapy aminosalicylate in comparison to UC patients who were treated with anti-TNF-α antibody or received no medication. All other sphingolipids did not differ between the various groups (Figure 5).

#### 3.3.3. Expression Level of Enzymes of the Sphingolipid De Novo Pathway

We checked CerS expression in white blood cells (WBC) from UC patients in comparison to WBC of the healthy control group. CerS1 and CerS3 mRNA expression significantly increased in WBC from UC patients in comparison to WBC from controls (Figure 6), whereas CerS4 mRNA expression significantly decreased. The expression level of CerS2, CerS5 and CerS6 do not differ significantly in WBC from UC patients in comparison to controls (Figure 6). These data correlate with the enhanced expression level of CerS1 and CerS3 in inflamed colon tissue (Figure 2) facilitating our hypothesis that the invasion of immune cells is possibly responsible for these changes. However, CerS expression in WBC did not correlate with the sphingolipid level in plasma. Further analyses are needed to specify the cells that are mainly responsible for the sphingolipid content in plasma and to identify the immune cells that mainly express CerS1 and/or CerS3 in colon tissue. 

#### 3.3.4. LC-QTOFMS Analysis of Lipids in Plasma

Untargeted analysis of blood lipids indicates that several other lipids such as lysophosphatidylcholines (LPC), triglycerides (TG) and free fatty acids (FA) significantly differ between UC patients in remission and with mild disease (Figure 7A). Interestingly, the precursor of resolvins (eicosapentaenoic acid (EPA) and docosahexaenoic acid (DHA)) significantly increased in blood of UC patients with disease stage (Figure 7B). Therefore, other blood lipid concentrations might be determined as possible biomarkers for disease control.

## 4. Discussion

Ten years ago, Duan and Nilsson highlighted the importance of sphingolipids in the gut for inflammatory bowel disease [36]. The high turnover rate of mucosal cells is associated with a high rate of lipid synthesis to ensure membrane integrity of the epithelium. Since it has been shown that dietary sphingomyelin (SM) does not contribute to the SM pool in atherogenic cholesterol and triglyceride-rich plasma lipoproteins and there is no evidence that the mucosa can uptake plasma lipoprotein sphingolipids from the blood, the sphingolipids for renewal of the colon epithelium must be primarily produced locally by the de novo synthesis [36].

These observations were confirmed by a recently published study showing that the serine palmitoyltrasferase (SPT), which is the rate-limiting enzyme in the sphingolipid de novo synthesis, is essential for intestine cell survival and barrier function [31]. Tamoxifen induced knock down of Sptlc2 in colon epithelial cells lead to diarrhea and rectal bleeding, ending in death after 7–10 days. Repression of SPT suppressed the level of ceramides and sphingomyelins in membranes of colon cells, which was accompanied by a reduction of E-cadherin and mucin2 and an induction of apoptosis in the colon. Disruption of the intestinal barrier function came along with an enhanced inflammation and bacterial invasion comparable to IBD in humans [31]. These data emphasize the importance of the sphingolipid de novo synthesis in the colon. Our data demonstrate that the sphingolipid de novo synthesis is repressed—especially in inflamed colon tissue from UC patients. This effect is not due to a reduced expression of Sptlc2/3 or CerS1-6. The human SPT is a heterodimer complex consisting of the subunits Sptlc1 and Sptlc2 or Sptlc3 and the composition might change dynamically in dependence of the tissue-specific expression of Sptlc2 and Sptlc3 [37,38]. However, because we saw no changes in the expression of Sptlc2 or Sptlc3 in our colon samples, we assume that inhibition of de novo synthesis might be related to the posttranscriptional regulation mechanism of SPT or CerS1-6 activity [23,39].

In inflamed colon tissue, we could detect a low, insignificant increase in CerS1 and CerS3 expression but this is likely related to the invasion of immune cells. This hypothesis was supported by an enhanced expression of CerS1 and CerS3 mRNA in WBC of UC patients in comparison to controls. Additionally, we could show that especially LacCer—but not GlcCer—significantly increased in inflamed colons in colitis patients. LacCer serve as pattern-recognition receptors (PRRs) on human cells to which several pathogenic microorganisms, including *Escherichia coli, Bordetella pertussis, Bacillus dysenteriae, Propionibacterium freudenreichii,* and *C. albicans,* bind and thereby activate an innate immune response [40,41]. Therefore, an increase in LacCer in inflamed colon tissue contributes to an enhanced binding of pathogens, thereby increasing immune response and inflammation.

However, many different mouse models have been used to investigate the influence of distinct sphingolipids in colitis. In CerS2-knockout (ko) mice, depletion of very-long chain ceramides is connected with disruption of the membrane integrity by downregulation of tight junction protein occludin and zonula occludens-1 (ZO-1), leading to enhanced severity of AOM/DSS-induced colitis in these mice [28]. CerS6-ko mice are protected from T-cell-mediated colitis but show severe symptoms in the DSS-evoked colitis model [29]. Inhibition of acid sphingomyelinase, which hydrolases sphingomyelin to ceramide, leads to an amelioration of DSS-induced colitis, possibly by inhibition of the immune cell response in this mouse model [30]. However, in our cohort we could only detect a significant decrease in the dihydroceramide level but no significant changes in the ceramide level in inflamed colon in comparison to control tissue. Therefore, our data indicate that in human colon tissue, inhibition of de novo synthesis, but not a disequilibrium between different ceramides, contributes to the development of colitis.

Several papers investigated the role of S1P in colitis. Inhibition of sphingosine kinase 1 (SK1), either by inhibitors or using SK1-ko mice, ameliorates DSS-induced colitis in mice [35,42], whereas downregulation of SK2 enhanced DSS-induced colitis in mice, which has been ascribed to a concomitant upregulation of SK1 in these mice [43]. In contrast, the degradation of S1P in colon epithelial cells by sphingosine-1-phosphate phosphatase 2 promotes disruption of mucosal integrity and enhances DSS-induced colitis [33]. Therefore, the effect of S1P seems to depend on local or systemic production. We determined S1P in blood and colon tissue of UC patients. However, due to the low amounts of S1P in colon tissue and the small tissue sample size, S1P was under the detection limit. In human blood, we detected a reduced S1P level in UC patients in comparison to controls. Our data show no changes in S1P level between UC patients in different stages; in addition, the common drugs used for the treatment of UC patients had no influence on the S1P level (Figure 5). Also in pediatric IBD patients, no significant changes in S1P serum level between the different disease stages could be detected [25]. The serum S1P level decreased within other inflammatory conditions such as pancreatitis [44], or septic shock patients [45,46]. However, the opposite has also been reported, as the plasma S1P level increased in community-acquired pneumonia [47] or in human papillomavirus (HBV)-genotype D positive patients [48]. These data indicate that the plasma S1P level can be up- or down-regulated under various inflammatory conditions and that further studies investigating where plasma S1P comes from are needed. Erythrocytes are one of the main S1P sources, but also binding of S1P to high-density lipoprotein or serum albumin might be an important mechanism that contributes to alterations in the serum S1P level [49]. However, even we and others could not detect a disease-dependent change in the plasma S1P level; manipulation of the S1P signaling pathway by treating UC patients with ozanimod seems to be a promising approach that several clinical trials are ongoing, investigating S1P modulators in IBD [50].

In contrast to S1P, we detected a significant increase of several sphingolipids in plasma from UC patients in comparison to control plasma. Interestingly, also in plasma from MS patients, C16:0-Cer, C24:1-Cer, C16:0-GlcCer and C24:1-GlcCer were up-regulated in comparison with healthy controls [51], which further strengthen their usability as biomarkers for disease control under inflammatory conditions [52]. Until now, it is not known where sphingolipids in plasma mainly come from. For S1P, it is known that it is produced and secreted from hematopoietic cells such as erythrocytes and endothelial cells [53]. Ceramides and other lipophilic sphingolipids are released as extracellular vesicles from different cells such as platelets, immune cells and from different tissues [54,55]. Further studies are needed to determine which cells are mainly responsible for the sphingolipid content in plasma in different diseases and to estimate their usefulness as biomarkers for disease control.

In addition to sphingolipids, we also detected other lipids in the blood of UC patients and revealed significant changes in free fatty acids (Figure 7). Saturated fatty acids can activate toll-like receptors, thereby enhancing inflammation, whereas unsaturated fatty acids counteract this mechanism [56,57]. DHA and EPA are the precursors of resolvins that are attributed to anti-inflammatory effects. Resolvins, such as RvE1-3 (derived from EPA) or RVD1-4, and maresins (derived from DHA) are synthesized endogenously and are important players in the resolution of colitis-associated inflammation [58,59,60,61]. Resolvins themselves are hard to detect in human blood samples but animal studies have shown that in a DSS-induced colitis mice model, the supplementation of fish oil enriched with omega-3 fatty acids such as EPA and DHA relieved the symptoms of colitis-associated inflammation by influencing redox-regulation and inhibition of pro-inflammatory processes [62,63]. In humans, it has been shown that supplementation with fish oil resulted in improvements in the histology index in UC patients [64]. The nutrition supplementation with EPA also has protective effects in patients with UC and is, therefore, suggested to be used as adjuvant to maintain symptom-free remission [65]. However, for patients taking these fatty acids as a dietary supplement, they may not be further useful as biomarkers for disease control.

## 5. Conclusions

Our study and literature data suggest that inhibition of the sphingolipid de novo synthesis in colon tissue might be a mechanism contributing to the development of UC (see also Figure A1). Furthermore, several sphingolipids and free fatty acids in the blood may be useful biomarkers for disease control, as they increase with severity. However, further studies are needed to investigate how useful these lipids are in detecting early changes in disease stages either in the direction of resolution or disease progression. Highly sensitive and selective detection methods have already been published [66] and are a prerequisite for further studies.

## Figures and Tables

**Figure 1 jcm-08-00971-f001:**
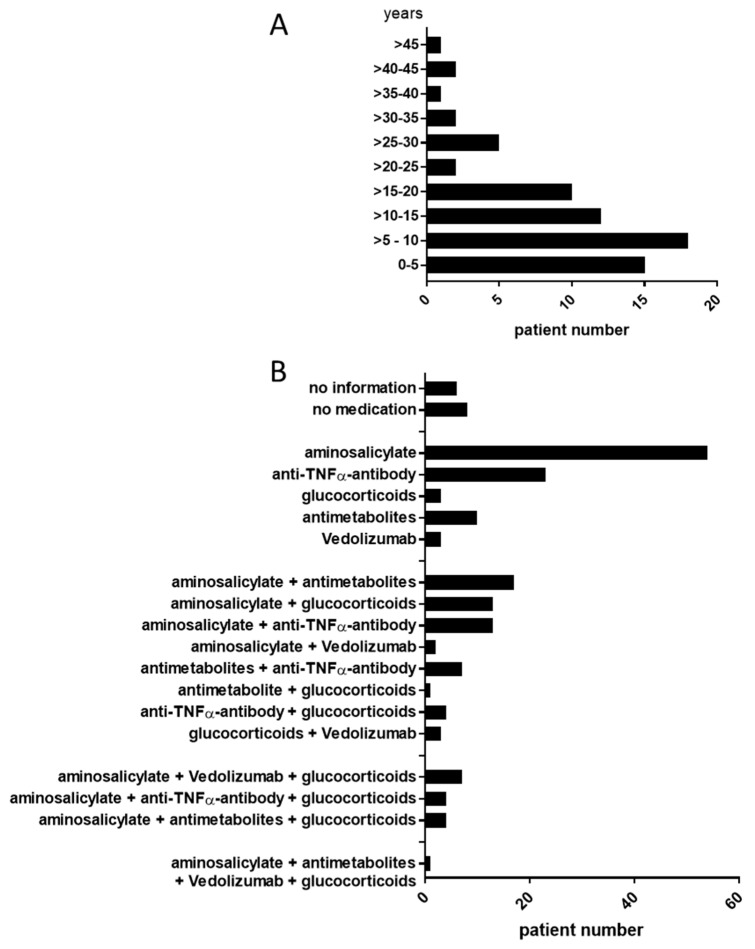
Patient data. (**A**) Time of disease duration in our ulcerative colitis (UC) patient cohort. (**B**) Overview of medical treatments of UC patients included in this study.

**Figure 2 jcm-08-00971-f002:**
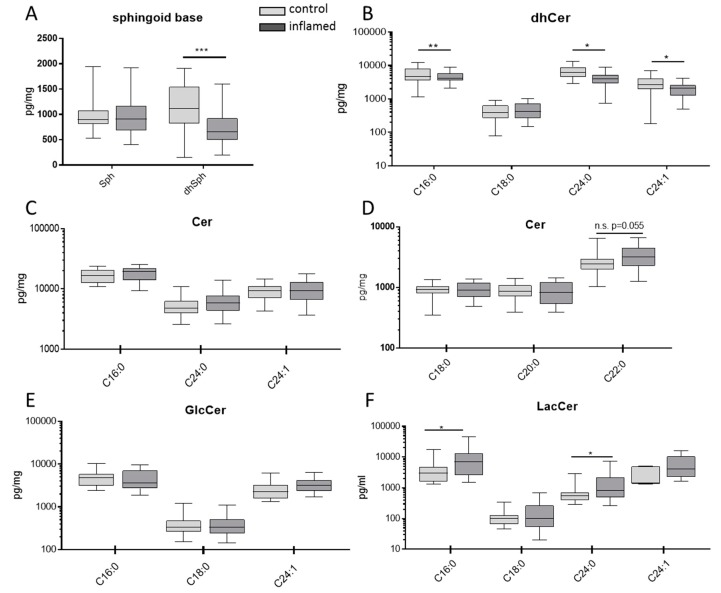
Sphingolipids in colon tissue. Colon samples from inflamed and control tissue were collected from UC patients undergoing diagnostic colonoscopy and sphingolipid concentrations where analyzed by LC–MS/MS. *n* = 27. Statistical differences were analyzed by a paired T-test, α = 0.05, (* *p* < 0.05, ** *p* < 0.01, *** *p* < 0.001).

**Figure 3 jcm-08-00971-f003:**
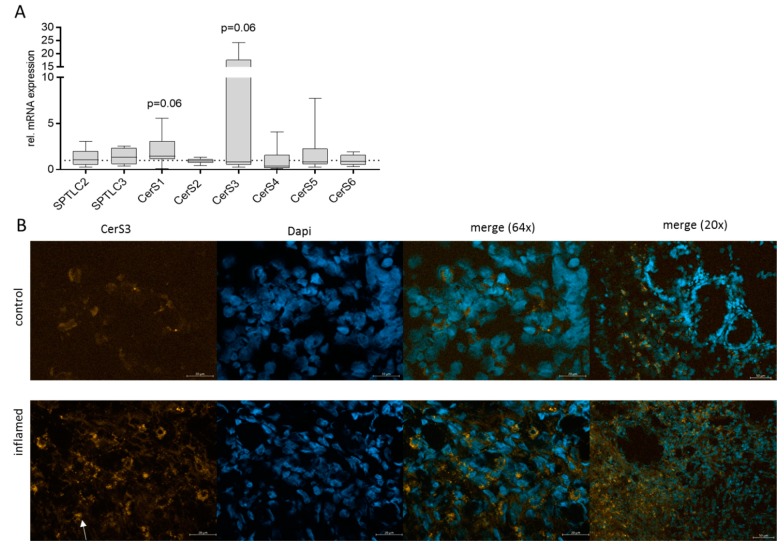
qRT-PCR and immunohistochemical staining of CerS and SPT in colon tissue. (**A**) mRNA levels of Sptlc2, Sptlc3 (SPT), CerS1-6 in inflamed and control tissue were detected by quantitative RT-PCR. CT levels were normalized to GAPDH and subsequently, related to the level of control (set to 1), using 2^(-ΔΔCT) method. Data are means ± SEM; *n* = 10–15. Statistical differences were analyzed by GraphPad Prism using unpaired T-test, *p* ≤ 0.05. (**B**) Immunohistochemistry of CerS3 in human colon tissue. CerS3 was stained with an anti-CerS3 antibody and a secondary antibody coupled to Cy3 and tissue was co-stained with DAPI. Original magnification: 63x and 20x; white arrow indicates cytoplasmic CerS3 staining in cell localized in the lamina propria.

**Figure 4 jcm-08-00971-f004:**
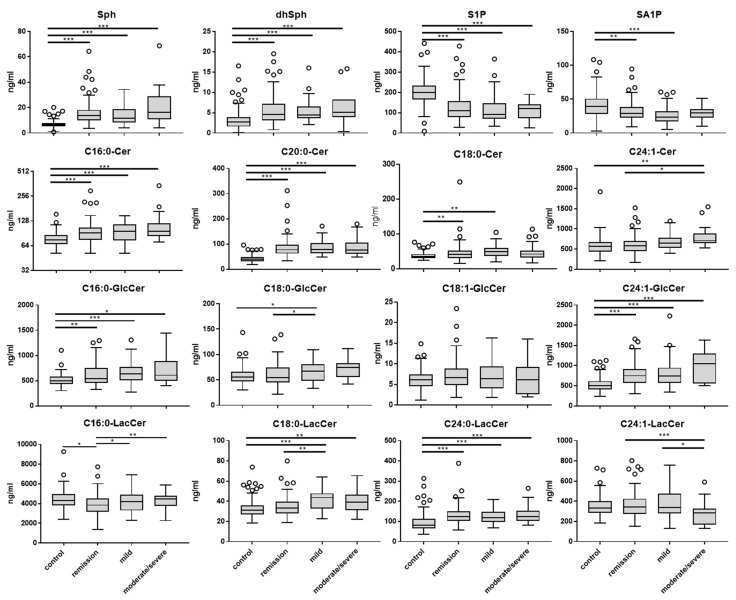
Sphingolipids in human plasma. Sphingolipids in human plasma was determined by LC–MS/MS. *n_control_* = 148; *n_remission_* = 102, *n_moderate_* = 43; *n_acute_* = 24, Statistical analysis was performed with R, significant differences between groups were assessed using two-tailed ANOVA with Tukey multiple comparison, α = 0.05. (* *p* < 0.05, ** *p* < 0.01, *** *p* < 0.001).

**Figure 5 jcm-08-00971-f005:**
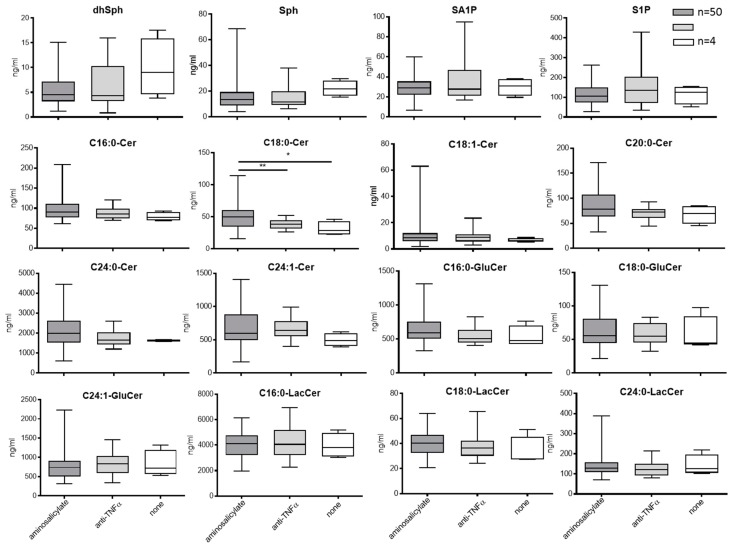
Sphingolipids in human plasma in relation to medical treatment. *n_none_* = 6; *n_anti TNF-α_* = 21; *n_aminosalicylate_* = 50. Statistical analysis was done with R, significant differences between groups were assessed using two-tailed ANOVA with Tukey multiple comparison, α = 0.05. (* *p* < 0.05, ** *p* < 0.01, *** *p* < 0.001).

**Figure 6 jcm-08-00971-f006:**
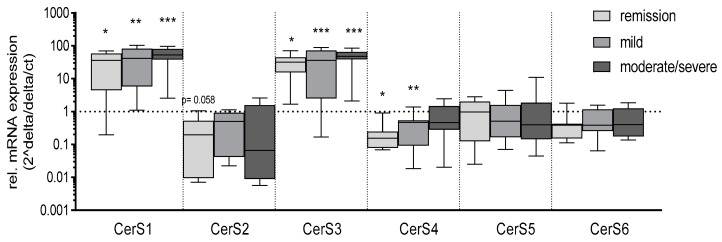
qRT-PCR of CerS in white blood cells from colitis patients and controls. mRNA levels of CerS1-6 in white blood cells (WBCs) were detected by quantitative RT-PCR. CT levels were normalized to GAPDH and subsequently related to the level of the control cohort (set to 1), using 2^(-ΔΔCT) method. Data are means ± SEM; *n* = 9–15. Statistical differences were analyzed by one-way ANOVA with α Bonferroni post-hoc test. α = 0.05. (* *p* < 0.05, ** *p* < 0.01, *** *p* < 0.001).

**Figure 7 jcm-08-00971-f007:**
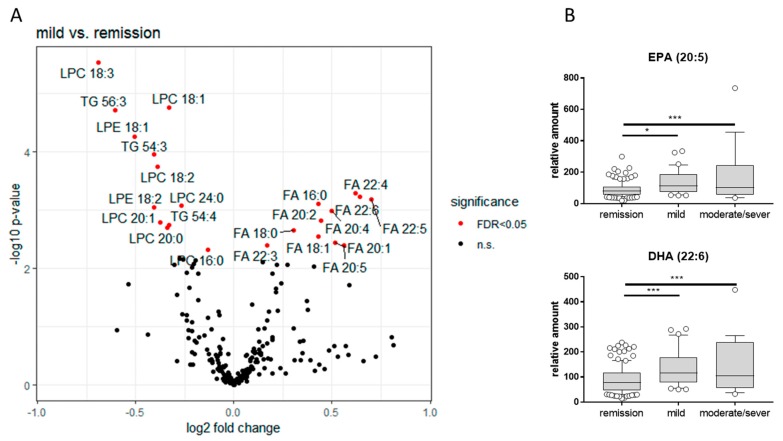
Lipids in human plasma. (**A**) Volcano blot of lipids in human plasma determined by LC-QTOFMS. Shown are the significances versus fold-changes of all lipids measured in plasma of UC patients who are in remission or suffer from mild disease. *n_remission_* = 109, *n_moderate_* = 33. Statistical analysis was performed with R, significant differences between groups were assessed using two-tailed ANOVA with Tukey multiple comparison, α = 0.05. (**B**) Relative amounts of EPA and DHA in plasma of UC patients with different disease stages. *n_remission_* = 109, *n_moderate_* = 33; *n_acute_* = 19. Statistical analysis was performed with GraphPad, significant differences between groups were assessed using two-tailed ANOVA with Tukey multiple comparison, α = 0.05. (* *p* < 0.05, ** *p* < 0.01, *** *p* < 0.001).

**Table 1 jcm-08-00971-t001:** qRT-PCR primer.

Gene	Primer Sequence	Manufacturer
Sptlc2	Fw 5′-TATGGAGCTGGAGTGTGCAG-3′ Rev 5′-GAATTCGTTGCAAATCCCAT-3′	biomers, Ulm Germany
Sptlc3	Fw 5′-GTATGATGAGTCTATGAGGAC-3′ Rev 5′-CATTCAGGAACTTAGCCACA-3′	biomers, Ulm Germany
CerS1	Fw 5′-CCTCCAGCCCAGAGAT-3′ Rev 5′-AGAAGGGGTAGTCGGTG-3′	biomers, Ulm, Germany
CerS2	Fw 5′-CCAGGTAGAGCGTTGGTT-3′ Rev 5′-CCAGGGTTTATCCACAATGAC-3′	biomers, Ulm, Germany
CerS3	Fw 5′-CCTGGCTGCTATTAGTCTGAT-3′ Rev 5′-TCACGAGGGTCCCACT-3′	biomers, Ulm, Germany
CerS4 (WBC)	Fw 5′-CTG GTG GTA CCT CTT GGA GC-3′ Rev 5′-CGT CGC ACA CTT CTA ATA CC-3′	biomers, Ulm, Germany
CerS4 (tissue)	Fw 5′-CTG GTG GTA CCT CTT GGA GC-3′ Rev 5′-AGC AAC ATC AGA AGC CCG TT-3′	biomers, Ulm, Germany
CerS5	Fw 5′-CAAGTATCAGCGGCTCTGT-3′ Rev 5′-ATTATCTCCCAACTCTCAAAGA-3′	biomers, Ulm, Germany
CerS6	Fw 5′-AAGCAACTGGACTGGGATGTT-3′ Rev 5′-AATCTGACTCCGTAGGTAAATACA-3′	biomers, Ulm, Germany
GAPDH	Fw 5′-CCA GGA GCG AGA TCC CTC-3′ Rev 5′-GGG CAG AGA TGA TGA CCC TT-3′	biomers, Ulm, Germany

**Table 2 jcm-08-00971-t002:** Patient data.

**Gender** **all patients** male female **colon-biopsis** male female **blood samples** male female	**number** 51/98 47/98 37/59 22/59 105/183 78/183	**(%)** (55) (45) (63) (37) (57) (43)
**Median age**	43.8 (19–76) years	
**Median disease duration**	11 (1–62) years	
**Disease status using partial mayo score (%)** remission mild moderate/severe	109/183 45/183 29/183	(60) (24) (16)
**Severity of disease (%)** pancolitis extensive colitis left side colitis sigmoiditis proktitis not mentioned	86/183 9/183 43/183 25/183 14/183 6/183	(47) (5) (23) (14) (8) (3)

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
