# Peer review of "The Lipid Status in Patients with Ulcerative Colitis: Sphingolipids are Disease-Dependent Regulated"

_jcm, 2019, doi:10.3390/jcm8070971_

Round 1
Reviewer 1 Report
There is an interesting papers presenting new data on sphingolipids in patients with ulcerative colitis, which should be a subject for further studies.
However, some improvement of the text is needed.
In the study 97 UC patients and 25 control subjects were enrolled. There is hard to understand why the number of blood samples is presented (183 and 148, respectively) as it is a technical problem.
Fig.3 – what do p values presented on Fig.3 mean? What do these values tell us?
The presented data only indicate that during further study new biomarker useful for disease monitoring could be recognized. The sample size is too small, the number of patients at the certain level of disease severity is small, and therefore, there is hard to accept that “ Several lipids changed significantly in blood which might be used as biomarkers for disease control. Dietary intake has an significant impact on plasma fatty acids, and EPA and DHA are often used (supplements) as adjuvant treatment for diseases associated with inflammation. The range of EPA and DHA observed in patients with moderate/severe disease is wide (Fig.7).
Author Response
We thank the reviewer for handling our paper and the helpful comments which we want to field as follows:
1) In the study 97 UC patients and 25 control subjects were enrolled. There is hard to understand why the number of blood samples is presented (183 and 148, respectively) as it is a technical problem.
We add an additional sentence in the Experimental Section (Page 3, line 104-107) explaining the background why the study contains more blood samples than patient numbers.
2) Fig.3 – what do p values presented on Fig.3 mean? What do these values tell us?
We show the p-values in Fig.3 because the data are only barely not significant. The p-values should give the reader a hint that there is only a tendency. Suchlike we described the results in the text (Page 12 line 297).
3)....there is hard to accept that “ Several lipids changed significantly in blood which might be used as biomarkers for disease control. Dietary intake has a significant impact on plasma fatty acids, and EPA and DHA are often used (supplements) as adjuvant treatment for diseases associated with inflammation. The range of EPA and DHA observed in patients with moderate/severe disease is wide (Fig.7).
We agree with the reviewer that further large studies are needed, to test the usefulness of the here identified lipids as biomarkers for disease control. Our study intended to find out which lipids might be useful biomarkers. Therefore, we added a sentence to the abstract where we pointed out that diet related variabilities have to be considered when lipids are used as possible biomarkers (Page 1 line 38 and Page 20 line 487)
Reviewer 2 Report
Bazarganipour S. et al presented here a very elegant study regarding the ulcerative colitis and sphingolipid. They have used lipidomics, RT-PCR and IHC for the investigation of several enzymes in the S1P pathway and concluded the study with two major changes of lipid- sphinganine(dhSph) and dyhydroceramides (dhCer) in the UC. The paper is very well drafted and nicely flow through and very easy to read. I just have a few comments to further improve the manuscript:
1. How did the lipidomic data normalize? Do the authors normalize each sample with protein concentration? In the manuscript Line: 175 authors state "Plasma samples 20uL" but didn't mentioned how did they normalize across samples?
2. Figure 3B didn't show specific immunofluorescent staining. The red staining seems to locate in the acellular area and can potentially be an artifact. Better images are needed.
3. Can the authors discuss the result in Fig4 panel 3 (S1P)? They found the S1p is significantly reduced compared with healthy control, which may have contradict to the general belief of S1p.
4. How did the human plasma lipidomic normalized? These details are essential.
Author Response
We thank this reviewer for handling our paper and the helpful comments made to our manuscript, which we willingly address in our revision:
1. How did the lipidomic data normalize? Do the authors normalize each sample with protein concentration? In the manuscript Line: 175 authors state "Plasma samples 20uL" but didn't mentioned how did they normalize across samples?
We added a paragraph to LC-QTOFMS experimental section where we described in detail how we normalized the samples to quality control samples and so to each other. Please see Page 7, line 219-227.
2. Figure 3B didn't show specific immunofluorescent staining. The red staining seems to locate in the acellular area and can potentially be an artifact. Better images are needed.
We repeated immunohistochemical staining with another antibody and show the new results in Fig. 3 with a better staining of CerS3, especially in cells localized in the lamina propria.
3. Can the authors discuss the result in Fig4 panel 3 (S1P)? They found the S1p is significantly reduced compared with healthy control, which may have contradict to the general belief of S1p.
We added a paragraph to the discussion addressing the here measured S1P level and the role of S1P signaling for IBD treatment.
Reviewer 3 Report
Manuscript ID: jcm-521916
Title: NUTRITIONAL REHABILITATION IN PATIENTS WITH MALNUTRITION FOR CROHN’S DISEASE
Authors : Sarah Bazarganipour, * Johannes Hausmann *, Stephanie Oertel, Khadija El-Hindi, Sebastian Brachtendorf, Irina Blumenstein, Alica Kubesch, Kathrin Sprinzl, Kerstin Birod, Lisa Hahnefeld, Sandra Trautmann, Dominique Thomas, Eva Herrmann, Gerd Geisslinger, Susanne Schiffmann, and Sabine Grösch
This is a prospective study where authors investigated sphingolipid and other lipids status in human plasma, white blood cells and 88 colon biopsies from UC patients. Analysis was also done based on disease severity and medical treatment in UC patients. They showed that similar to others studies, inhibition of sphingolipid de novo synthesis in colon tissue is a contributing factor in the development of the disease.
This study is very well written and has extensive data to support the author’s results and conclusion. The only comment I have is about the immunohistochemical staining. The images provided are not convincing and look like nonspecific staining. Perhaps a better image or enlarging the area/ pointing at the areas they want to highlight might help. Otherwise, everything looks good.
Author Response
We thank the reviewer for the kind comments and address the comments as follows:
1) The only comment I have is about the immunohistochemical staining. The images provided are not convincing and look like nonspecific staining. Perhaps a better image or enlarging the area/ pointing at the areas they want to highlight might help.
We repeated immunohistochemical staining with another antibody and show the new results in Fig. 3 with a better staining of CerS3, especially in cells localized in the lamina propria.
Round 2
Reviewer 2 Report
Greatly improved from previous version. Very nice work.